# Cone-Traced Supersampling for Signed Distance Field Rendering

Andrei Chubarau*
McGill University

Yangyang Zhao†
Huawei Technologies
Canada

Ruby Rao‡
Huawei Technologies
Canada

Paul G. Kry§
McGill University

Derek Nowrouzezahrai¶
McGill University

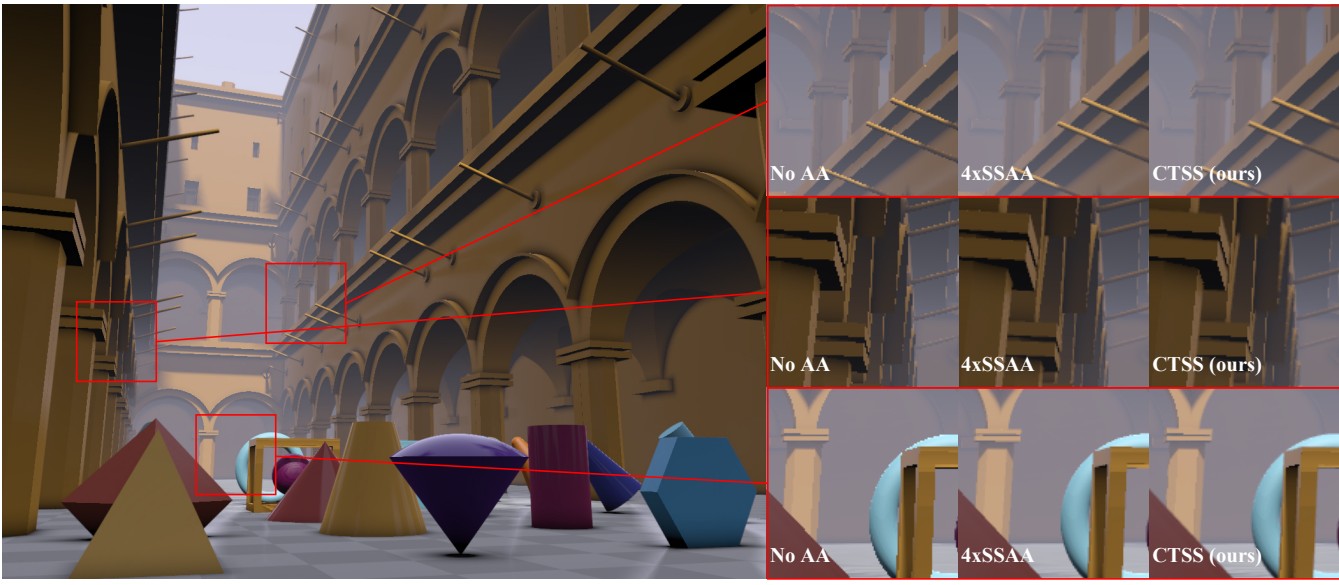

Figure 1: Different antialiasing techniques for SDF-based scene rendering. The full scene shown on the left is rendered at 720p with our antialiasing method (CTSS). The zoomed-in patches compare the original non-antialiased image against renders with antialiasing achieved using 4x supersampling antialiasing (SSAA) and CTSS. While 4xSSAA quadruples rendering time, antialiasing with CTSS incurs a significantly lower performance cost ($\approx 1.3\times$) and generally provides similar edge smoothing and overall visual quality for most geometric edges.

## ABSTRACT

While Signed Distance Fields (SDFs) in theory offer infinite level of detail, they are typically rendered using the sphere tracing algorithm at finite resolutions, which causes the common rasterized image synthesis problem of aliasing. Most existing optimized antialiasing solutions rely on polygon mesh representations; SDF-based geometry can only be directly antialiased with the computationally expensive supersampling or with post-processing filters that often lead to undesirable blurriness and ghosting. In this work, we present cone-traced supersampling (CTSS)[1], an efficient and robust spatial antialiasing solution that naturally complements the sphere tracing algorithm, does not require casting additional rays per pixel or offline pre-filtering, and can be easily implemented in existing real-time SDF renderers. CTSS performs supersampling along the traced ray near surfaces with partial visibility identified by evaluating cone intersections within a pixel's view frustum. We further devise a specialized sampling strategy to minimize the number of shading computations and aggregate the collected samples based on their correlated visibility. Depending on configuration, CTSS incurs roughly 15-30% added computational cost and significantly outperforms conventional supersampling approaches while offering comparative antialiasing and visual image quality for most geometric edges.

*e-mail: andrei.chubarau@mail.mcgill.ca
†e-mail: yangyang.zhao2@huawei.com
‡e-mail: ruby.rao@huawei.com
§e-mail: kry@cs.mcgill.ca
¶e-mail: derek@cim.mcgill.ca

**Index Terms:** Computing methodologies—Computer graphics—Image manipulation—Antialiasing; Computing methodologies—Computer graphics—Shape modeling—Parametric curve and surface models; Computing methodologies—Computer graphics—Rendering—Ray tracing

## 1 INTRODUCTION

Whereas conventional polygon meshes cannot fully represent curved surfaces, Signed Distance Fields (SDFs) provide a continuous and smooth implicit representation of a surface [41]. Various mathematical properties of SDFs have been actively studied in computer graphics and vision applications [14,20]. SDF-based representations may be used to facilitate certain real-time geometric calculations and manipulations [17,31,38,48]; renderers may even fully omit conventional 3D modelling in favor of SDF-based representations [37,40]. Furthermore, as SDFs offer guaranteed continuity and differentiability [41], they are naturally applicable to deep learning for tasks involving shape representation and reconstruction [13,26,42,47].

Instead of explicitly defining the geometric structure of an object, SDFs encode the distance to the boundary of a given shape for any point in space [41]; the sign further indicates whether the current point is inside or outside the object. Visualization of SDF-based objects requires modifications to the conventional rendering pipelines since most graphical renderers are designed for polygon mesh geometry. Rendering SDFs is thus typically done with sphere tracing [21], an algorithm that iteratively converges along a given ray towards the zero-value iso-surface of an SDF – the boundary of the associated shape – thereby enabling its shading and further processing.

---

[1]Interactive CTSS demo and code on Shadertoy: https://www.shadertoy.com/view/7lSXWK

Because modern renderers commonly rely on rasterization to sample and shade object surfaces within a scene at a predefined grid-like pattern of pixels – regardless of the underlying shape representation – rendering is prone to aliasing artifacts due to undersampling. Representing a pixel by a single discrete point sample limits the ability to resolve subpixel features and variation, leading to visually jagged appearance as well as spatial and temporal noise when combined with motion. Visible aliasing artifacts are a significant detriment to perceived image quality of the rendered imagery: antialiasing is thus one of the fundamental challenges in real-time computer graphics.

Many common antialiasing solutions rely on variants of supersampling to aggregate multiple samples per pixel thus reducing the visibility of aliasing artifacts at the cost of added computational overhead [1, 9, 56]. However, due to the popularity of conventional 3D representations, most of the optimized antialiasing methods are designed for mesh-based geometry, preventing their application for rendering SDFs. Post-processing [33, 44] and temporal filters [54], on the other hand, often result in blurriness or ghosting artifacts with lower overall image quality when compared to the expensive but high-quality supersampling.

As SDFs naturally facilitate evaluation of cone intersections, the original work on sphere tracing [21] foresees the theoretical application of pre-filtered cone tracing for antialiasing in SDF-based pipelines. Practical implementations, however, indicate that this comes at the cost of substantial offline preprocessing, additional memory requirements, and significantly reduced performance [2, 19, 23, 49]. Nevertheless, cone tracing is a powerful general tool in graphics: cones have a wider receptive field and can approximate the entire view frustum or more accurately describe the footprint of a pixel. Partial cone intersections further allow to identify local proximity to surfaces, which can be leveraged for various graphical effects, such as soft shadows and global illumination [11] as well as screen-space reflections [24].

In this paper, we address antialiasing specifically in the context of rendering SDFs and introduce Cone-Traced Supersampling (CTSS), a practical cone-tracing-based spatial antialiasing method that naturally complements the sphere tracing algorithm with no pre-filtering and minimal computational overhead. Unlike conventional supersampling which naively collects multiple samples for every pixel, CTSS performs contextual supersampling along the traced ray: we collect additional shaded color samples near object surfaces as detected by partial cone intersections within the given pixel's footprint. We further introduce a specialized sampling strategy to minimize the number of collected samples, optimizing the performance of our method, and aggregate collected samples by resolving sub-pixel visibility to keep track of sample correlation, which we find is necessary for consistent antialiasing. Compared to a baseline implementation of sphere tracing, CTSS incurs approximately 15-30% reduction in framerate, significantly outperforming conventional SDF-based antialiasing methods while providing high-quality antialiasing comparable to $4\times$-$8\times$ supersampling for most geometric edges.

The rest of this document is structured as follows: in Sect. 2, we provide an overview of the relevant literature focusing on SDF-based representations and rendering as well as various antialiasing techniques. Sect. 3 covers our proposed antialiasing solution, cone-traced supersampling (CTSS), including optimization and implementation details. We provide empirical results in Sect. 4, comparing CTSS to other available antialiasing solutions for SDF rendering in terms of visual quality and performance. We conclude the paper in Sect. 5 with a discussion of limitations and possible future improvements.

## 2 RELATED WORK

A brief overview of relevant literature is provided, with a primary focus on Signed Distance Fields and antialiasing methods.

**Signed Distance Fields.** Unlike explicit models that directly describe the geometry of 3D objects, implicit surfaces are typically defined by the iso-contour of a level-set function $f : \mathbb{R}^3 \to \mathbb{R}$. For SDFs, an object is described by encoding the distance to the closest point on the object's surface [41], such that the implicit shape corresponds to the region defined as $\{\mathbf{x} \in \mathbb{R}^3 : f(\mathbf{x}) \leq 0\}$. Points inside the object are thus characterized by negative SDF values, while the zero-value SDF iso-surface is the object's boundary. Compared to meshes and point clouds, SDFs can describe continuous surfaces with arbitrary resolution and varying topologies. SDFs can also be easily expressed as voxel fields [12, 38], occupancy maps [36], and even directly encoded with neural networks [13, 42, 47].

**Rendering SDFs.** To visualize an SDF-based object, the implicit surface can first be polygonized [14, 32, 41] allowing the use of existing rendering techniques at the expense of accuracy and large memory and computational overhead. Alternatively, SDFs can be more accurately rendered in real-time using sphere tracing [21], an iterative algorithm that asymptotically converges along a particular ray to the zero isosurface of the SDF function.

**Sphere Tracing.** Similarly to ray tracing, the ultimate goal of sphere tracing is to determine if a ray intersects with a given shape. Ray-SDF intersections occur along the zero-value isosurface of the SDF. The sphere tracing algorithm asymptotically approaches the surface by advancing along a given ray with step distance equal to the current SDF value, i.e., the radius of an *unbounding sphere* [22]. This guarantees no immediate intersection with the implicit surface and no over-stepping, with eventual convergence to the zero-value boundary of the SDF function $f$. Mathematically, sphere tracing can be expressed iteratively as

$$t_{i+1} = t_i + SDF(\boldsymbol{p}_t), \qquad (1)$$

where $t_i$ and $t_{i+1}$ correspond to the current and next depth values along a given ray, respectively, while $\boldsymbol{p}_t$ is the point in space associated with $t_i$ defined as

$$\boldsymbol{p}_t = \boldsymbol{r}_o + \boldsymbol{r}_d t_i, \qquad (2)$$

given the traced ray with origin $\boldsymbol{r}_o$ and direction $\boldsymbol{r}_d$.

SDFs provide solely the distance to the object and not the direction to the corresponding point; this results in asymptotic convergence, approaching but never truly reaching the surface. In practical implementations, intersections occur when the value of the distance function $SDF(\boldsymbol{p}_t)$ falls below some predefined threshold $\epsilon$. Sphere tracing thus often involves hundreds of steps to reach the threshold value and each iteration requires evaluating the SDF at the current position. Furthermore, SDF values naturally decrease when nearing a surface thereby reducing the step size and further slowing down the convergence.

Compared to classical ray tracing which casts infinitesimally thin rays from the viewpoint, the sphere tracing algorithm is more similar to beam or spherical cone tracing as it terminates with a finite controllable threshold. Moreover, the footprint of a pixel is more accurately described using beams or cones, which would normally require casting multiple rays (e.g., supersampling, Monte-Carlo methods [45]). Lastly, while cone tracing can become prohibitively complex for mesh-based representations with thousands of primitive polygons, it is by design trivial with SDFs.

**Antialiasing Methods.** Although the simplistic supersampling antialiasing (SSAA) can reduce the adverse effect of aliasing, it incurs large computational and memory costs hindering its application to real-time rendering; a multitude of specialized antialiasing techniques have thus been developed to address the limitations of SSAA. For instance, high frequency detail can be prefiltered for textures [39, 51], specular highlights [28], and shadows [43], and then more accurately sampled according to the pixel's footprint. Alternatively, with hidden surface removal mechanisms providing a

more efficient method to resolve sub-pixel visibility [6, 7], modern graphical pipelines typically decouple the expensive shading operations from the relatively cheap geometric coverage sampling [56] and improve on SSAA by using variants of multi-sampling antialiasing (MSAA) [1] that leverage various strategies to further reduce the number of shading calculations [8, 10, 16, 46, 50]. However, as such methods operate with polygon primitives, they are not applicable for rendering SDFs.

Antialiasing can also be achieved via image enhancement and post-processing filters [27, 33, 34, 44]: gradient discontinuities or various special patterns can be detected in an image and the problematic areas can be smoothed out via blurring to provide a less jagged appearance. Temporal antialiasing and reconstruction methods [29, 35, 52, 54, 55] further improve the stability and quality of the antialiased image by leveraging a history buffer of the past frames to filter out noise and gather additional geometric and color information. With the recent popularity of deep learning, antialiasing was also successfully achieved with neural networks via real-time super-resolution [15] and neural upsampling [53], although such methods tend to require proprietary hardware and setup complexity.

In the context of rendering SDFs, it may also be relevant to consider sphere tracing acceleration techniques as these may indirectly contribute to antialiasing by reducing the computational cost of computing additional samples per pixel, i.e., via cheaper supersampling. Typically, such methods contextually increase the step size of sphere tracing in an attempt to speed up convergence to the surface and thus to minimize the number of SDF evaluations, which are the most computationally demanding component [3, 4, 18, 30]. Since these methods specifically concern accelerating sphere tracing and not antialiasing, we do not explore them in this work.

## 3  CONE-TRACED SUPER SAMPLING

We term our antialiasing solution Cone-Traced SuperSampling (CTSS) as our method relies on detecting cone intersections and acquiring multiple shaded color samples within the view frustum of a given pixel. Conventional sphere tracing evaluates a single intersection per pixel. Supersampling further sub-divides the pixel into $k$ independent sphere tracing instances. CTSS, on the other hand, collects up to $k$ samples per pixel along the original ray and correlates sample visibility by keeping track of accumulated occlusion. We evaluate cone intersections – this is trivial with SDFs as it only requires comparing the current SDF value to the cone radius at the current ray depth – to identity surfaces with partial visibility, which are then sampled to resolve sub-pixel detail and achieve antialiasing. Moreover, instead of terminating at the first intersection with a surface, we adaptively control the minimum step size for sphere tracing to continue iterating until full cone occlusion is detected to ensure that all visible surfaces are identified.

In what follows, we will describe our modifications to the conventional sphere tracing algorithm to evaluate partial geometric intersections within the footprint of a given pixel, our efficient sampling strategy to minimize the number of shading computations, and how we compute the final antialiased color by aggregating the collected samples based on correlated surface visibility.

### 3.1  Cone Tracing for CTSS

While we reuse the overall iterative mechanism of sphere tracing, CTSS additionally evaluates cone intersections along the traced ray. We define the size of the traced cone such that its circular base passes through the four corners of a given pixel to ensure full pixel coverage. Cone tracing and CTSS computations only require $\tan(\theta)$, where $\theta$ is the half cone angle. Assuming that the vertical number of pixels corresponds to unit distance in camera space and given the viewport's vertical resolution $I_y$, we express $\tan(\theta)$ as the ratio

between half pixel size $d_p$ and the camera's focal length $F$:

$$\tan(\theta) = \frac{d_p}{F} = \frac{\sqrt{2}}{F \times I_y}. \tag{3}$$

For cones with small angle $\theta$, cone radius $R_c$ can be quite accurately approximated for any ray depth $t$ as

$$R_c = t \times \tan(\theta). \tag{4}$$

Due to the effect of perspective in rectangular viewports, pixels at the center of the image have slightly wider cone coverage than pixels on the extremities; this is mainly noticeable for cameras with small focal length and lower resolutions. However, for most practical setups, we find that ignoring the effect of perspective is an acceptable approximation. That is, we treat all pixels as if they were aligned along the central view direction. We can further easily modify the coverage of a cone by applying a constant multiplier to the property $\tan(\theta)$ or directly $R_c$, i.e., we can enforce cone coverage to cover more than a single pixel. Note that $R_c$ is diagonal across a pixel; pixel coverage must be scaled by a factor of $\sqrt{2}$ when converting to cone radius.

Lastly, because SDF values approach zero or can turn negative near the surface – this prevents further progression along the ray and slows down the convergence of sphere tracing – in order to continue tracing past a surface, we must enforce a lower bound on the step size. The step size should be small enough to guarantee adequate intersection detection, but large enough to escape a hard hit. To satisfy this condition, we compute an adaptive lower bound for the step size as $s_{min} = \alpha R_c$, where $\alpha < 1$ and $R_c$ is the current cone radius. For CTSS, we set $\alpha = 0.5$, which offers a balance of speed and accuracy. We describe our strategy to correct intersection detection to account for non-optimal step sizes in Sect. 3.6.

### 3.2  Cone Occlusion

It is convenient to describe cone intersections in terms of cone occlusion, i.e., an estimate of how much of the traced cone is occupied by a given geometric intersection. Given the SDF value $d$ and the cone radius $R_c$ for the current iteration of sphere tracing, we compute cone occlusion $\omega$ similarly to the method presented by Ban et al. [5]:

$$\omega = \frac{1 - d/R_c}{2}. \tag{5}$$

We graphically illustrate cone occlusion in Fig. 2. Negative $\omega$ values occur when $R_c < d$, corresponding to no cone intersection. Positive cone occlusion implies $d < R_c$, indicating that a cone intersection is detected. Full cone coverage occurs inside the surface when $d < -R_c$ (SDF is negative inside the object). We term these different types of cone intersections as *soft* ($0 < \omega$) and *hard* ($0.5 < \omega$) hits. While the original sphere tracing algorithm terminates when the first hard hit is detected along the traced ray, i.e., $d < \epsilon$ for some small threshold $\epsilon$ (using cone occlusion, $0.5 - \epsilon < \omega$), CTSS continues cone tracing until full cone occlusion ($1.0 - \epsilon < \omega$) to ensure that all visible surfaces are encountered.

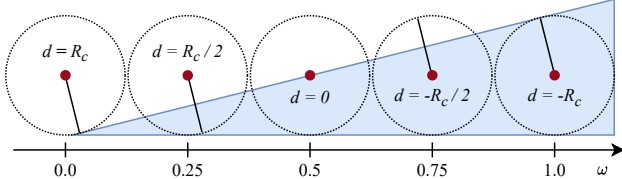

Figure 2: Cone intersections with varying cone occlusion. The traced surface is shown in blue; the cones, as circular cross sections. The ratio of SDF value $d$ to cone radius $R_c$ is directly correlated with cone occlusion $\omega$. Adapted from [5].

## 3.3 Sampling Strategy

CTSS analyzes sphere tracing iterations with cone intersections to select where to collect shading samples. Naively, we could sample the local color at each iteration with non-zero cone occlusion, but as sphere tracing may require hundreds of iterations to converge to the surface and partial intersections with geometry may occur at any iteration, this is largely impractical due to computational constraints. It is also necessary to avoid oversampling the same surface multiple times as this may cause potentially biased and inconsistent results.

To avoid the computationally expensive iteration-based sampling, we first identity hit *groups*, defined as a set of subsequent iterations with a continuous cone intersection. Hit groups tend to naturally split the traced ray into segments. A given hit group $g_i$ has an *entry* point at ray depth $t_{in}$ – the first position for the associated cone intersection – and may have an *exit* point at $t_{out}$. While the ray segment associated with a hit group $g_i$ is not exactly on the surface of the evaluated SDF, it effectively approximates the local surface and can be actively sampled. In theory, the visible representation of a hit group can be computed as an integral of all visible colors from the associated surface; in practice, we find that a single color sample is in most cases sufficient to describe an entire hit group. Moreover, since not all points on the surface within $g_i$ are visible to the observer due to local occlusion and normal orientation, it is critical to position the sample adequately.

The instinctive solution is to sample at the average position of a given hit group $g_i$ or at the point with the smallest SDF value (locally maximal cone occlusion), i.e., somewhere between $t_{in}$ and $t_{out}$. However, we find that this simple approach often results in shading the surface at grazing angles or, even worse, with the surface normal oriented away from the observer. Instead, we opt to shade at the entry point of a given hit group $g_i$, i.e., at the associated ray length $t_{in}$. The local geometry at the entry point ensures that the surface will be oriented towards the observer, thus avoiding the problem of backface shading. If a hard hit is detected for a particular group, the sample is collected at the position of the first hard hit in $g_i$ and not the entry point. To avoid double sampling soft hits that become hard hits, we sample the entry point $t_{in}$ only once the exit point $t_{out}$ is identified, indicating that the trace is exiting proximity with a surface and thus not leading to a hard hit.

We illustrate the geometric configuration of our sampling scheme for a generic trace in Fig. 3. In this example, three soft hit groups are detected within the tested cone; the second and the third groups both contain a hard hit, and the trace terminates with full cone occlusion in the third group. As per our strategy, CTSS will collect three shaded color samples indicated on the x-axis of the plot: the entry point of the first soft hit group at $t_{1in}$, the hard hit in the second group at $t_{2h}$, and the final hard hit in the third group at $t_{3h}$.

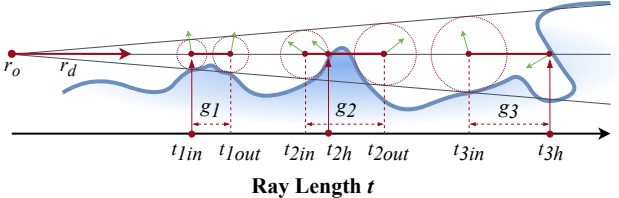

**Ray Length t**

Figure 3: Geometric properties of cone-traced supersampling along the traced ray and for a given surface (in blue). Soft hit groups $g_i$ are illustrated as red segments along the central ray, with the normal vectors at relevant points shown in green; the sample points recorded by CTSS are indicated as red circles on the x-axis. Note that the second group contains a hard hit at $t_{2h}$, but CTSS continues tracing until full cone occlusion is detected at $t_{3h}$.

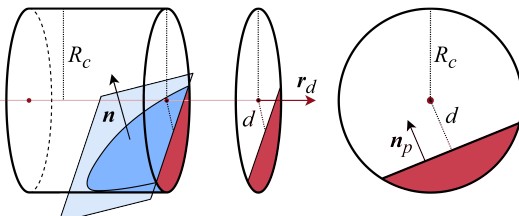

Figure 4: Macroscopic model of cone-SDF intersections. A segment of the traced cone is visualized (left) as a cylinder; the intersecting surface defined by the SDF is assumed to be macroscopically planar. Projected onto the pixel footprint, the surface is seen (right) as a half-space, thus describing its visible coverage within the pixel. Adapted from [23].

## 3.4 Subpixel Surface Visibility

Not all surfaces detected within the traced cone are equally visible from the original view point. Geometry may be partially occluded by previously encountered surfaces or due to local occlusion within a hit group. The detected hit groups should then accordingly have unequal visual contributions to the final observed color.

With CTSS, we resolve sub-pixel visibility to allocate weights for the collected samples. Similar to differential cone-tracing presented by Heitz and Neyret [23], we estimate the visible contribution of a given surface by computing its relative occupancy in the pixel's footprint. Heitz and Neyret assume that the macroscopic curvature of the surface is negligible, hence the surface can be locally approximated as a plane. As illustrated in Fig. 4, the cone-SDF intersection is modelled from the perspective of the camera, i.e., along the 2D circular cross-section of the traced cone representing the pixel. Given the surface normal $\vec{n}$ projected on the pixel footprint to obtain $\vec{n}_p$, the SDF value $d$, and the cone radius $R_c$, the surface in the cross-section of the cone is represented as a 2D half-space defined by $\vec{n}_p$ and a distance $d$ away from the center of the cone. As the plane naturally divides the pixel into two regions, we can test against the half-space to compute a 2D surface visibility mask over the pixel.

Borrowing from differential cone-tracing, which likewise takes inspiration from the A-Buffer originally described by Carpenter [6], we pack the visibility mask in the bits of a single 32-bit integer. The visibility of a surface is encoded in the number of ones in the bitmask; surface clipping computations become simple boolean operations. We can thus efficiently correlate the visibility between subsequent hit groups by keeping track of occlusion using a global visibility bitmask $M$ updated after each sample. The mask $M$ is initialized with all zeros to indicate zero occlusion; for each new sample, we compute the associated visibility bitmask $m_i$, test against $M$ to remove invisible bits, and lastly, update $M$ with the newly occluded bits before moving on to the next sample.

The final antialiased color $C$ is computed as the weighted sum

$$C = \frac{\sum_i w_i C_i}{\sum_i w_i}, \tag{6}$$

given the shaded color contributions $C_i$ of the detected hit groups and the corresponding visibility-based weights $0 < w_i < 1$, where $w_i = bitCount(m_i)/32$. With this modification, we effectively increase the resolution of the trace by considering sub-pixel visibility, which largely benefits the visual quality of the final antialiased image. We illustrate the associated visual improvement in Fig. 5 (c) and (e) , while (b) and (d) demonstrate that simply averaging the samples with uniform weights results in a visual effect similar to edge extrusion with the original aliasing. The difference is even clearer when the cone coverage of CTSS is manually increased to multiple pixels.

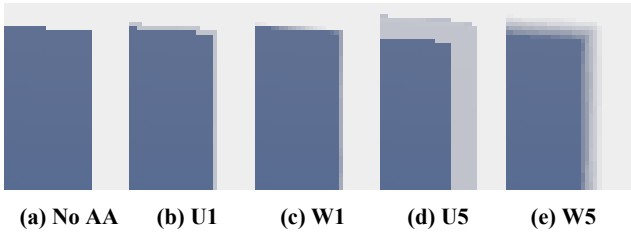

**(a) No AA**    **(b) U1**    **(c) W1**    **(d) U5**    **(e) W5**

Figure 5: CTSS sample aggregation using uniform **U** and visibility-based weights **W**. The surface (blue) with no antialiasing is shown in (a) as reference; CTSS with 1 pixel cone coverage shown in (b) and (c), and CTSS with 5 pixel coverage in (d) and (e).

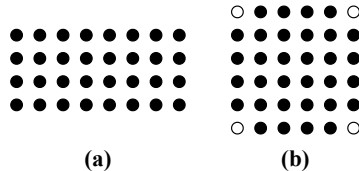

**(a)**        **(b)**

Figure 6: Visibility bitmask encoded in a 32-bit integer as (a) $4 \times 8$ mask from A-Buffer [6], (b) $6 \times 6$ mask with omitted corners (ours).

### 3.5 Visibility Bitmask Implementation

Unlike the original $4 \times 8$ bitmask encoding sub-pixel visibility used by differential cone-tracing and the A-Buffer, we use a $6 \times 6$ bitmask with the corner bits omitted to more closely match the circular cross-section of the traced cone (see Fig. 6). Moreover, while Heitz and Neyret pre-compute and store all possible visibility bitmasks in a look-up table, we compute local visibility masks in real-time. Our bitmask implementation has insignificant computational overhead, but eliminates the need for additional textures. We define the visible region with a line of the form $y = ax + b$ in the local frame of the cross-section of the traced cone, with the slope $a$ and y-intercept $b$ expressed as

$$a = -n_x/n_y, \tag{7}$$
$$b = d(n_x^2 + n_y^2)/n_y, \tag{8}$$

given projected normal $\vec{n}_p = (n_x, n_y)$ and the local SDF value $d$. Note that to avoid overflow and division by zero when computing $a$ and $b$, we clamp the minimum absolute values in $\vec{n}_p$.

Pixel footprint is given by the traced cone with radius $R_c$, hence the mask spans $2R_c$. We solve for the relative positions $0 < x_i < 2R_c$ over six horizontal lines corresponding to the six rows of our $6 \times 6$ bitmask. The positions $x_i$ are then converted to the number of required bit-shifts to apply to a full 111111 bitmask to encode the desired coverage, and the orientation of coverage is given by $\vec{n}_p$. For example, 000111 corresponds to half coverage from the right with $x_i = R_c$, and requires 111111 to be shifted right by three bits; coverage from the left implies left bit shifts.

### 3.6 Backtracing Correction

Regular sphere tracing accurately converges to the ray-SDF intersection since the step size is equal to the local SDF value. However, if the step size has a lower bound – which is the case for CTSS – the safe stepping distance with no intersection is no longer guaranteed near the surface, hence hard hit detection becomes less accurate. In addition, the entry and exit points for cone intersections are always determined with spatial delay regardless of the step size, as illustrated in Fig. 7.

CTSS samples shading information at the positions of hard hits and cone hit entry points. Sampling delayed hits may cause visual artifacts and inconsistencies, e.g., shading inside the surface or at surfaces not facing the observer. To more accurately represent

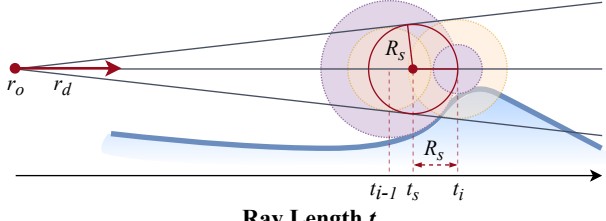

**Ray Length *t***

Figure 7: Backtracing at the cone intersection entry point. Since sphere tracing detects a "delayed" cone intersection at $t_i$, the true cone intersection entry point occurs earlier, between $t_{i-1}$ and $t_i$. Regions tested for intersection are shown as purple circles (radius equals local SDF value); orange semi-transparent circles show the associated cone radii; solid red circle depicts our *safe* cone radius and the corrected intersection entry point after backtracing.

intersections with local geometry, we apply "backtracing" to step backwards along the traced ray by a safe fallback distance that ensures no intersection. Given ray depth values $t_{i-1}$ and $t_i$ for two subsequent sphere tracing iterations with a cone intersection at $t_i$ but not at $t_{i-1}$, we estimate a *safe* ray depth $t_s$ such that $t_{i-1} < t_s < t_i$.

For hard hits, we must correct intersection detection due to non-optimal step size for sphere tracing with CTSS, i.e., $d_{i-1} \leq t_i - t_{i-1}$, where $d_{i-1}$ is the SDF value at previous iteration. We simply return to the previously determined safe ray depth $t_s = t_{i-1} + d_{i-1}$.

For cone intersections, we define $t_s$ in terms of a *safe* cone radius $R_s = t_i - t_s$. We reuse Equation 4 to express $R_s$ in terms of the cone coverage angle $\theta$ as

$$R_s = t_s \times \tan(\theta), \tag{9}$$

and then compute $t_s$ as

$$t_s = \frac{t_i}{1 + \tan(\theta)}. \tag{10}$$

The geometry of our *safe* cone with the circular base located at ray depth $t_s$ along the traced ray ensures that no cone intersection occurs with the SDF and provides a more reliable entry point estimate during cone tracing. Backtracing correction is also applied to the intersection exit point. The involved calculations are very simple with negligible performance cost, while the corrected positions produce much more reliable shading samples.

## 4 EXPERIMENTS AND RESULTS

We tested various resolutions and scene complexities representative of popular workloads to evaluate the performance and the robustness of CTSS, and compare our antialiasing solution to sphere tracing with supersampling (SSAA) in terms of image quality and computational load. All our tests were performed on a Windows 10 machine with an Intel® Core™ i7-11900K CPU and NVIDIA GeForce RTX 3090 GPU. We implemented sphere tracing with and without CTSS in Shadertoy[1] [25] (GLSL) for exploration and in Unity Engine [48] (HLSL) for performance evaluations. We also tested a variant of CTSS with relaxed termination criteria (CTSS-R), which terminates at the first hard hit (half cone occlusion), offering faster antialiasing but lower image quality. For SSAA, we sample on a uniform $k \times k$ grid for each pixel (e.g., $2 \times 2$ for 4xSSAA).

Since the overall performance, accuracy, and final rendered image given by sphere tracing largely depend on the value of the SDF threshold used to determine intersections with the geometry, it was necessary to set equivalent thresholds for sphere tracing with and without CTSS. To ensure consistent comparison, we implemented sphere tracing with an adaptive cone-based threshold for intersection detection. This small modification uses a termination threshold that increases with ray depth instead of a constant value.

Table 1: Performance evaluation in various scenes and resolutions comparing no antialiasing, supersampling, and CTSS. For each method, we report the frametime (FT) in milliseconds as well as the ratio against the reference performance with no antialiasing.

| Scene | Primitives | | | | Random Box | | | | Random Shape | | | | Sponza SDF[1] | | | | Sparse Shape | | | | |
|---|---|---|---|---|---|---|---|---|---|---|---|---|---|---|---|---|---|---|---|---|---|
| Resolution | 1280x720 | | 2560x1440 | | 1280x720 | | 2560x1440 | | 1280x720 | | 2560x1440 | | 1280x720 | | 2560x1440 | | 1280x720 | | 2560x1440 | | |
| Method | FT (ms) | Ratio | FT (ms) | Ratio | FT (ms) | Ratio | FT (ms) | Ratio | FT (ms) | Ratio | FT (ms) | Ratio | FT (ms) | Ratio | FT (ms) | Ratio | FT (ms) | Ratio | FT (ms) | Ratio | Avg Ratio |
| No AA | 4.67 | 1.0 | 17.1 | 1.0 | 7.07 | 1.0 | 25.14 | 1.0 | 14.59 | 1.0 | 47.58 | 1.0 | 6.62 | 1.0 | 25.82 | 1.0 | 8.67 | 1.0 | 23.67 | 1.0 | 1.0 |
| 4x SS | 20.82 | 4.46 | 66.25 | 3.87 | 29.47 | 4.17 | 103.5 | 4.12 | 61.16 | 4.19 | 192.2 | 4.04 | 28.38 | 4.29 | 106.42 | 4.12 | 36.82 | 4.25 | 96.82 | 4.09 | 4.16 |
| CTSS | 6.08 | 1.30 | 20.38 | 1.19 | 9.72 | 1.37 | 34.78 | 1.38 | 19.21 | 1.32 | 59.73 | 1.26 | 9.04 | 1.37 | 31.96 | 1.24 | 11.08 | 1.28 | 31.08 | 1.31 | 1.30 |
| CTSS-R | 5.15 | 1.10 | 17.98 | 1.05 | 9.12 | 1.29 | 32.41 | 1.29 | 17.92 | 1.23 | 57.24 | 1.20 | 6.96 | 1.05 | 27.78 | 1.08 | 10.02 | 1.16 | 28.02 | 1.18 | 1.16 |

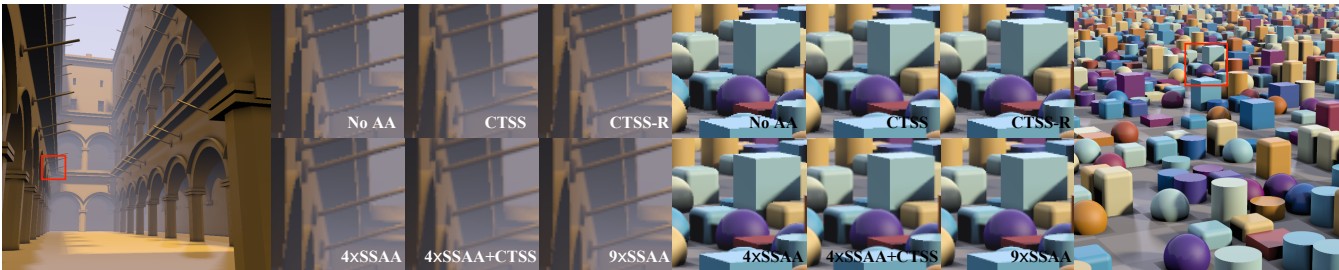

Figure 8: Image quality comparison between CTSS and supersampling (SSAA). We showcase CTSS renders at 720p for two scenes: Sponza SDF and Random Shape. SSAA collects multiple samples for every pixel resulting in uniform antialiasing quality. CTSS performs supersampling specifically near object boundaries but not for flat surfaces, textures, or edges that it cannot detect. As a result, CTSS provides comparable visual quality near object edges but is overall less consistent; CTSS-R provides a marginally smaller visual improvement. Combining CTSS and supersampling yields superior quality at fewer samples per pixel as demonstrated by 4xSSAA + CTSS.

We summarize the pseudocode for sphere tracing with CTSS in Algorithm 1. Note several simplifications for clarity, for instance, we omit definitions for readily available logical tests – e.g., for hit entry and exit. We similarly omit certain configurable parameters.

### 4.1 Performance Analysis

Table 1 presents our performance assessment for various scenes and resolutions. The tests were run using our HLSL implementation for sphere tracing and CTSS in Unity Engine; we used Unity's *Profiler* and *Profile Analyzer* tools, and report the mean frametime for 600 subsequent frames. Averaging the results from all our tested scenes, CTSS contributes to roughly 30% overhead, while CTSS-R is even more efficient incurring only 16% slowdown. CTSS scales well for scenes with different geometric complexity, and produces consistent performance results under various configurations.

While supersampling approaches essentially multiply the required amount of work by the supersampling factor – e.g., 4xSSAA quadruples the computational load – CTSS results in a marginal increase in computation correlated with the number of complex pixels in the rendered image, significantly improving the performance of antialiasing. Because CTSS extends sphere tracing with supersampling and collects multiple color samples along the traced ray, it is naturally slower than sphere tracing itself (no AA). CTSS contains additional logic and increases the number of shading operations per pixel. Fur-

thermore, sphere tracing slows down near object boundaries due to small step sizes; as CTSS continues sphere tracing until full cone occlusion is detected, it may encounter multiple surface boundaries, slowing down multiple times. CTSS-R eliminates the possibility of recurrent proximity to surfaces leading to better performance.

### 4.2 Antialiasing Quality Analysis

We qualitatively analyze the image quality of the five tested scenes rendered with different antialiasing solutions and showcase two of the fives scenes in Fig. 8. Compared to uniform supersampling, CTSS specifically targets spatial regions near edges, as partial cone intersections naturally occur at grazing angles in proximity with geometry; CTSS does not antialias textures or flat geometry as no edges are detected in these regions. We further illustrate the visual difference between CTSS and CTSS-R in Fig. 9. CTSS-R softens the appearance for pixels near the object's boundary but does not directly antialias pixels with existing hard hits; antialiasing is achieved by smoothing the extruded edge. The full implementation of CTSS, on the other hand, traces until full pixel occlusion and applies antialiasing to both sides of the edge leading to improved visual quality. Lastly, SSAA and CTSS can be combined for improved visual quality by supersampling in three dimensions: in pixel space (2D) plus along the traced ray (3D). For instance, 4xSSAA and CTSS provide visual quality comparable to 9xSSAA at less than half the cost. Combining antialiasing in ray-space (CTSS) and image-space (supersampling) thus yields the best visual quality.

### 5 DISCUSSION

In this section we describe the key configurable CTSS parameters that influence the visual quality and performance, provide a brief discussion of limitations, and highlight directions for future work.

### 5.1 CTSS Configurations

**Cone Coverage.** The standard CTSS implementation with cone coverage set to a single pixel can be modified to use larger cones for intersection detection: cone radius $R_c$, defined in Equation 4, can be

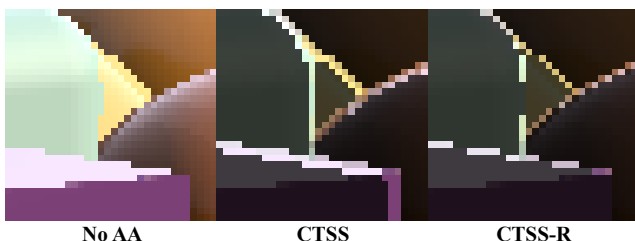

No AA       CTSS       CTSS-R

Figure 9: Affected pixels for CTSS and CTSS-R. Complex pixels (highlighted) occur at object contours. Full CTSS antialiases more pixels as it continues tracing until full cone occlusion, potentially encountering multiple hit groups. CTSS-R stops at the first hard hit, detects fewer hit groups, and thus antialiases fewer pixels.

---

[1]Sponza SDF from `https://github.com/mmerchante/sdf-gen-unity`

**Algorithm 1:** Pseudocode for weighted CTSS

**Input:** Ray origin $r_o$, ray direction $r_d$, SDF function $f$, tangent of cone angle *tanTheta*, maximum ray traversal distance $D$
**Output:** Antialiased color for a given pixel

```
1  t = t_i = 0                  // current and sample ray depths
2  M = 0                        // global visibility mask
3  ω_max = 0                    // keep track of max occlusion
4  c = (0,0,0)                  // accumulates pixel color
5  w_t = 0                      // total sample weight
6  while t < D do
7      p = r_o + tr_d           // current position
8      d = f(p)                 // SDF evaluation
9      R_c = t × tanTheta       // current cone radius
10     ω = (1 - d/R_c)/2        // cone occlusion metric
11     if soft hit then
12         ω_max = max(ω_max, ω)       // max occlusion
13         if (soft hit entry) or (first hard hit entry ) then
14             t_i = t          // update hit sample position

15     if (soft hit exit) or (full cone occlusion) then
           /* sample current group */
16         m_i = visibilityMask()      // sample visibility
17         m_i = m_i ∧ !M              // visibility correlation
18         w_i = bitCount(m_i)/32      // sample weight
19         c = c + w_i × shade(r_o, r_d, t_i)   // update color
20         w_t = w_t + w_i             // update total weight
21         M = M ∨ m_i                 // update global mask
22     if full cone occlusion then
23         break                // exit the loop, stop tracing
       /* always advance at least by R_c/2 */
24     t = t + max(d, R_c/2)
25 return c/w_t                 // normalize by total weight
```

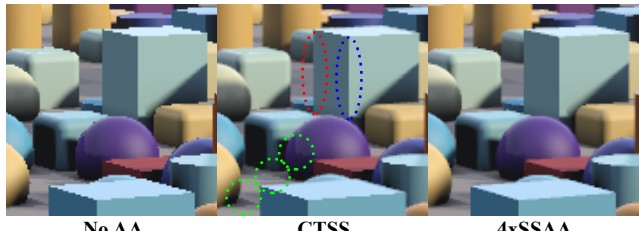

| No AA | CTSS | 4xSSAA |
|-------|------|--------|

Figure 10: Limitations of antialiasing with CTSS. In red, correctly antialiased *external* edge; in blue, *internal* edge undetectable by our sampling strategy; in green (left to right), non-antialized color discontinuities from shadows, floor texture (checkerboard), and specular/reflection. CTSS performs geometric antialiasing at object contours, but does not antialias flat regions.

**Shading Calculations.** Shading can be computed either as soon as the hit groups are detected, or delayed until sphere tracing with CTSS is completed. With delayed sampling, it is necessary to record a list of ray depth values $t_i$ and the associated visibility masks $m_i$ during tracing. We term these two variants *ASAP* and *Delayed*, respectively. The ASAP strategy is most straightforward and requires the least amount of additional logic. The delayed variant allows to decouple coverage and shading but requires more memory. It also allows for computation in separate passes, i.e., running all coverage and all shading computations separately. We did not implement delayed sampling for CTSS, but it may offer further performance optimizations. This property can be leveraged using multiple passes in shader implementations. It may also be relevant when using neural SDFs (e.g., [13, 42]) as batching multiple SDF computations is then desirable for adequate performance.

**Visibility Threshold.** CTSS continues tracing until full cone occlusion or full bitmask visibility. Given our 32-bit integer mask, we can also opt to terminate tracing when the visibility mask has some set number of $k < 32$ bits filled. Best antialiasing quality is achieved with $k = 32$ but smaller visibility thresholds offer a minor speed up. This is demonstrated with CTSS-R terminating at half cone occlusion, which is similar to $k = 16$.

## 5.2 Limitations of CTSS

**Antialiasing.** CTSS performs geometric antialiasing by detecting and sampling distinct hit groups during sphere tracing. We identify hit groups by their entry and exit points, and allocate a single shaded color sample per group. As a result, CTSS is incapable of resolving color discontinuities caused by textures, normals, and specular highlights that may occur within a single hit group. For instance, Fig. 9 highlights pixels where CTSS collects multiple samples: while the depth discontinuities near object contours – *external* edges – are detected and correctly antialiased, regions with relatively continuous depth – *internal* edges – cannot be identified. Similarly, flat regions with color variation cannot be detected under our sampling scheme. These properties are further illustrated in Fig. 10. Lastly, given our sampling strategy based on hit groups and our assumption that the macroscopic curvature of the surface is negligible, CTSS cannot accurately represent surfaces that are significantly smaller than the pixel size.

**Applicability.** While sphere tracing can non-optimally evaluate a function $f$ that is not a true SDF but satisfies $f(p) \leq SDF(p)$, CTSS may produce inconsistent antialiasing when rendering $f$. When $f$ underestimates the true distance to the object's surface, cone intersection detection is inconsistent and CTSS cannot correctly compute the visibility of the corresponding surface within a pixel. Although CTSS works well for piece-wise functions (as demonstrated by the highly complex Sponza SDF scene in Fig. 1 and Fig. 8), visual artifacts may be present when CTSS is applied to non-Eikonal functions where $\|\nabla f(\boldsymbol{x})\| \neq 1$.

multiplied by a constant. We observe that the visual smoothness of edges naturally increases with cone coverage wider than one pixel. Since CTSS-R only applies outer edge extrusion and smoothing – limiting the visible antialiasing effect – increasing its cone coverage improves the produced image quality. CTSS-R-2p (with two pixel coverage) then more closely matches the visual smoothing achieved with CTSS-1p. Note that expanding cone coverage leads to slightly reduced performance as more samples are collected and may cause visual artifacts for larger coverage values.

**Minimum Step Size.** The value for minimum step size (see Sect. 3.1) during CTSS is critical for performance. Smaller minimum values result in higher intersection detection accuracy but severely impact render time. We find that the step size of at least $R_c/2$ provides an adequate trade-off between accuracy and performance. Note that the minimum step size should be at most $R_c$ in order to not step over hard hits as well as to conform to our backtracing correction strategy.

**Number of Samples.** For typical scene complexities, CTSS detects anywhere between one to eight hit groups per pixel. Limiting the maximum number of collected samples is nevertheless a practical constraint. That being said, we identify little performance gain associated with limiting the number of samples per pixel as this is naturally achieved with cone tracing and our sampling scheme. Lastly, we note that coverage and shading computations may be decoupled, i.e., record sampling positions while tracing, but delay shading until later. Limiting the number of samples then reduces the memory requirements as fewer sample positions need to be recorded.

## 5.3 Future Work

While internal edges and color discontinuities are problematic for antialiasing with CTSS, an interesting direction for future work is to improve our sampling scheme optimization. More specifically, we may contextually collect multiple samples per hit group based on the group's local geometric properties, i.e., to further subdivide the hit group. One such possible detection strategy could examine the surface normals at different scales to identify distinct surfaces within a single group. Alternatively, to avoid potentially unnecessary normal computations, we could also simply look at the length to depth ratio of a hit group and allocate additional samples accordingly. Despite the additional computational cost of more sophisticated sampling strategies, we believe these costs can likely be justified for addressing the antialiasing limitations described above.

CTSS may further allow for rendering SDF objects with transparency. Since our modification to sphere tracing limits the minimum step size, we continue sphere tracing past a surface intersection, thus possibly encountering multiple surfaces. In the context of transparent objects, we may use a similar sphere tracing modification to correctly evaluate multiple surfaces per pixel. While CTSS currently uses a visibility bitmask to describe the local contribution of a given surface to the final pixel color, we can extend this to accumulate non-binary visibility thus considering transparent objects. For instance, this can be achieved by using $K$ bitmasks to encode $K$ levels of transparency.

## 6 CONCLUSION

We present cone-traced supersampling (CTSS), an efficient antialiasing solution for SDF-based rendering that leverages cone tracing to perform supersampling along the traced ray without casting additional rays per pixel or offline pre-filtering. Our method scales well to various levels of scene complexity, offers consistent image quality comparative to $4\times$ - $8\times$ supersampling for most geometric edges at a fractional computational cost, and is simple to implement on top of conventional sphere tracing. These advantages make CTSS an attractive method for a variety of real-time SDF rendering applications.

### ACKNOWLEDGMENTS

This work was made possible by the first author's internship at Huawei Technologies Canada.

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
