# OpenReview forum: "Cone-Traced Supersampling for Signed Distance Field Rendering"
_graphicsinterface.org/Graphics_Interface/2023/Conference_SD — GI 2023 - second deadline_

### Official Review · Reviewer_7ot8 · 2023-04-20
**An antialiasing paper that can address only a small subset of edges**

**Rating:** 6
**Confidence:** 3

**Review:**

The paper solves an antialiasing problem in a rendered image, a well-known but old research topic. Nevertheless, it targets a specialized rendering, i.e., a sphere tracing for a scene represented with SDFs, and presents an efficient and effective antialiasing solution.

The paper is written well, and the presentation is quite clear. In addition, the demonstrated results seem practical. I think the computational overhead (15 - 20% over the baseline without any antialiasing) is small, and the quality improvement is noticeable for some edges with geometric discontinuities.

Nonetheless, this aliasing seems only effective for the edges where the depth varies significantly (like object contours) and appears less beneficial for the borders with normal discontinuities. One can notice this behavior of this new work from Fig. 7 (e.g., inner edges in the boxes). I think it would be better to point out this limitation more explicitly in the Antialiasing paragraph in Sec. 5.2 so that a reader can immediately understand the practical scope of this antialiasing solution, i.e., which kinds of edges can be antialiased by this new work. For example, giving some examples (e.g., the edges shown in Fig. 7) would be helpful for a reader since this is not the case for the flat surfaces and texture discontinuities the paper mentioned.

In short, I enjoy reading this interesting paper, and the results seem practical with a small computational overhead. While it has a major limitation (i.e., only effective for the edges with depth discontinuities), I am positive about the paper.

---

### Official Review · Reviewer_z6XE · 2023-04-21
**Leaning towards acceptance**

**Rating:** 7
**Confidence:** 3

**Review:**

This paper tackles the problem of antialiasing in SDF rendering. They key idea is based on the cone-tracing and accumulate contributions from the partial hit between the cone and the underlying geometry.

In general, I am leaning towards accepting the submission. I find the overall idea elegant and effective. It leads to approximately 3x speed up compared to 3x supersampling while maintaining a similar rendering quality. The paper is also well-written and discuss reasons behind their design and assumptions. One minor suggestion to improve clarity is to consider separating some implementation details from the method section.

Despite leaning positive, I still have a couple questions/concerns:
1. In Sec. 3.4 the authors stated that "the surface can be locally approximated as a plane". I wonder whether there are cases where this assumption limits the object this method can render. For instance, would this assumption affects the rendering of thin objects such as hair or leaves?

2. I am also interested in seeing the failure case related to the non-SDF issue mentioned in  Sec. 5.2 - Antialiasing. As in practice, there are many non-SDF examples. For instance, CSG objects are Eikonal but not a true SDF. Neural SDFs / rasterizing SDF to a grid would also break the Eikonal 1 property. It would be beneficial to understand more about the failure cases among these "SDF-like" cases.

---

### Official Review · Reviewer_17pT · 2023-04-23
**Review - A lightweight AA method for SDF sphere tracing**

**Rating:** 8
**Confidence:** 3

**Review:**

**Summary:**
This paper addresses anti-aliasing of signed distance functions rendered with the sphere tracing algorithm.
Unlike conventional super sampling AA that traces multiple rays through a pixel, the core of the method borrows from differential cone tracing to gather and combine multiple surface samples along a single ray.
1. A cone covering one or more pixels is traced against the scene.
2. The cone / SDF intersection sections are found using sphere tracing and backtracking along the ray.
3. The colors ot the *soft* (cone) and *hard* (ray) hits are determined.
4. Each color is weighted using a visibility bitmask, that is updated at each sample with an approximate estimate of the cone occlusion.
5. The final pixel value is the weighted average of the surface samples.

The authors propose two variant of their algorithm one for high quality and one that trades quality for performance by stopping at the first hard hit encountered. The technique speed up the AA of sphere traced SDF compared to SSAA.

**Quality of References:**
They seem adequate, given my limited knowledge of the domain.
The differential cone method [23] has been proposed for efficient filtering of SVO data approximated as SDF elements.
Cone/SDF intersections have been used for soft shading in sphere tracing and in subsequent papers [21][5].
The authors borrow ideas from both of these methods and adress problem-specific issues to build their AA solution.
In my eyes, the presented method seems original and valuable.

**Reproducibility:**
All relevant information is given in the text and the source code is also provided (Unity package).
However, the source code for ShaderToy mentioned in the text is not and would have been appreciated.

**Clarity concerns:**
- Lot of real-world scenes include sem-transparent objects, hence a quick discusion about transparent surfaces would be appreciated.
    - Could that method work in presence of transparent objects ? If yes how, if not why?
- We can see from the top inserts in Figure 1 that the edges of the arch do not benefit from the CTSS method and look similar to the "No AA" method.
    - Could you please elaborate more on that matter? because CTSS is supposed to detect such edges.
- In the approach, the local visibility bitmasks are computed in real-time, whereas Heitz and Neyret precompute bitmasks in a LUT for performance reasons.
    - Would it be beneficial to have such a precomputed LUT in CTSS to reduce bitmask computations? If so, how can this be done?

**Further suggestions:**
- A figure and/or the pseudocode describing the local visibility mask computation would help reproducibility (i.e. the `visibilityMask()` function in the pseudocode).
- The presentation of the pseudocode could be improved for better readability.

**Explanation of Recommendation:**
The manuscript is in my eyes of good quality.
The problem is well motivated, the related work is described comprehensively and the method is positioned among the related work.
The figures and tables are self explanatory and clearly illustrate the appraoch and results.
The method is described in detail and is well reproducible - especially since source code is provided.
The method is evaluated on multiple scenes, and is compared to previous approaches.
In my eyes, this work is a valuable contribution and of high quality.

**Cons:** Other than the few suggestions listed above, I have no reservations about this article.